

# Short-term effects of thinning on the development and communities of understory vegetation of Chinese fir plantations in Southeastern China

Xuelei Xu[1], Xinjie Wang[1], Yang Hu[1], Ping Wang[2], Sajjad Saeed[3] and Yujun Sun[1]

[1] State Forestry Administration Key Laboratory of Forest Resources & Environmental Management, College of Forestry, Beijing Forestry University, Beijing, China
[2] College of Grassland Science, Beijing Forestry University, Beijing, China
[3] Department of Forestry & Wildlife Management, University of Haripur, Haripur, Pakistan

Corresponding author
Yujun Sun, sunyj@bjfu.edu.cn

## ABSTRACT

**Background**. High-density conditions are global issues that threaten the sustainable management of plantations throughout the world. Monocultures and untimely management practices have identically resulted in the simplex of community structures, decreases in biodiversity, and long-term productivity losses in plantations China. The most popular measure which is commonly used to address these issues is thinning, which potentially results in increases in the development of understory plants in plantations. However, there is limited information currently available regarding the community composition of understory vegetation and the associated environmental factors, which has limited the sustainable management of China's fir plantation ecosystems.

**Method**. In the present study, a thinning experiment was implemented which included a control check (CK: no thinning), as well as low intensity thinning (LIT: 20%), moderate intensity thinning (MIT: 33%), and high intensity thinning (HIT: 50%) in Chinese fir plantations located in the Southeastern China. During the investigation process, the understory vegetation examined three years after thinning measures were completed, in order to analyze the impacts of different thinning intensities on the growth and community composition of the understory plants. At the same time, the associated environmental factors in the fir plantations were also investigated.

**Results**. The species richness, total coverage, and biomass of the understory vegetation were observed to be apparently increased with increasing thinning intensity. In addition, it was found that the thinning measures had prominently influenced the soil nutrients. The community compositions of the understory vegetation were significantly different among the four thinning intensity levels, especially between the CK and the HIT. Furthermore, the development of the understory vegetation was found to be significantly correlated with the soil nutrient contents, and the community compositions of the understory vegetation were prominently driven by the tree densities, slope positions, and soil nutrient contents.

## INTRODUCTION

Over the past few decades, the Chinese government has extensively cultivated fast-growing trees in order to satisfy the increasing requirements for wood production (*Zhou et al., 2016*). It was estimated that the total area of tree plantations in China was approximately 69 million ha, which accounted for 24.64% of the global area (*SFA, 2014*; *FAO, 2015*). Chinese fir (*Cunninghamia lanceolata*) is an important tree species which has been widely planted in Southern China. It is known to be a type of fast-growing tree species, with high production levels and outstanding timber quality. It was estimated that the total area of Chinese fir plantations in China was approximately 8.95 million ha (*SFA, 2014*). However, it has been found that the stand volume and timber production have not been satisfactory, even though the areas of the plantations continue to increase (*Sheng, 2018*). This is due to the monocultures and high-density management practices which are in place at the present time (*Tian et al., 2011*). On one hand, the plantation forests, especially the pure coniferous plantations, were clearly recognized as having potential problems associated with monocultures, such as simplex community structures, low biodiversity, and a lack of biological stability (*Savill et al., 1997*). On the other hand, the majority of the plantations were initially planted with high densities and have not been efficiently managed (*Lindgren & Sullivan, 2013*; *Raunikar et al., 2010*; *Zhou et al., 2016*), which has resulted in the poor development of understory vegetation, delays in litter decomposition, and losses in essential soil nutrients (*Ma et al., 2007*; *Sheng, 2001a*; *Tian et al., 2011*). These issues have in turn brought lower long-term productivity and timber yields, and ultimately have seriously affected the sustainable management of the plantations (*Ares, Neill & Puettmann, 2010*; *Sheng, 2001b*; *Wen, Cheng & Liu, 2008*). However, the demand for emphasizing the development and community composition of understory vegetation during forest management practice in China has not been effectively supported (*Gilliam, 2007*; *Trentini et al., 2017*).

At present, thinning measures have been recognized as a practical method to resolve the aforementioned problems in Chinese fir plantations (*Sheng, 2001b*). Thinning measures are the most popular silvicultural treatment throughout the world in plantation forests, and are extensively used to increase the growth of reserved trees, as well as improve the productivity levels of plantations (*Trentini et al., 2017*). In addition, this type of management method directly or indirectly influences the forest characteristics (*Taki et al., 2010*), such as the understory vegetation, soil properties, and even the microorganisms within the forest ecosystems themselves (*Chen et al., 2015*; *Dang et al., 2018*; *Taki et al., 2010*). It has been found that the implementation of thinning practices improved the growth spaces and light transmittance in stands by reducing the canopy densities, and ultimately enhanced the growth of both the reserved trees and understory plants (*Ares, Neill & Puettmann, 2010*; *Cheng, Yu & Wu, 2013*; *Verschuyl et al., 2011*). Previous studies have extensively surveyed the impacts of thinning on the diversity and biomass of understory vegetation (*Cheng et al., 2014*; *Dang et al., 2018*; *Tamura & Yamane, 2017*; *Wen, Cheng & Liu, 2008*), as well as the effects on tree growth rates (*Gong, Niu & Mu, 2015*; *Wu et al., 2015*), physical and chemical characteristics of the soil (*Wang et al., 2013*; *Zhou et al., 2016*), and even the soil microbial

biomass (*Kim et al., 2018*; *Thibodeau et al., 2000*), nutrient transformations (*Boerner et al., 2008*), and soil respiration characteristics (*Akburak & Makineci, 2015*; *Tang et al., 2005*).

The understory vegetation is an important part of forest ecosystems and plays critical roles in their structures and functions. For example, it has been found that healthy understory vegetation could effectively increase biodiversity, form or improve community structures, and enhance the stability of forest ecosystems (*Baba et al., 2011*; *Harmer et al., 2016*; *Sheng, 2018*; *Yao, Sheng & Xiong, 1991*). A growing number of forest managers and researchers have begun to focus on the importance of understory plants in plantation management processes (*Torras & Saura, 2008*; *Brunet, Fritz & Richnau, 2010*; *Trentini et al., 2017*), in order to preserve the biodiversity and facilitate the formation and development of community structures, as well as maintain the various functions and long-term productivity of plantation ecosystems. Traditionally, various studies of thinning have provided inconsistent results on understory because focused on different tree species and conducted on different areas (*Nagai & Yoshida, 2006*; *Cheng et al., 2017*; *Lindh & Muir, 2004*). Particularly, some studies of thinning on Chinese fir plantations supported that the species richness, diversity, coverage and biomass of the understory plants increased with increasing thinning intensity (*Xiong, Sheng & Zeng, 1995*; *Wang, Li & Wang, 2010*; *Wang, Olatunji & Xiao, 2019*; *Zhou et al., 2016*). In contrary, another study showed that thinning have no effects on the diversity of understory in Chinese fir plantations (*Cheng et al., 2014*). The results of these previous studies have been far from universal and unified. In addition, a previous study demonstrated that plantation changed the community structures of understory compared with natural forests (*Piwczyński, Puchałka & Ulrich, 2016*). However, it remains unclear how thinning practices actually affect understory community structures and distribution patterns in plantation forest ecosystems.

Therefore, this study examined the responses of the diversity, coverage, biomass and community structure of understory vegetation to thinning and associated environmental factors in Chinese fir plantations located in Southeastern China, which is crucial to deepen the understanding of the effects of thinning on the basic processes of Chinese fir plantation ecosystems in order to guide the future management of such systems. We hypothesized that the thinning intensities would significantly affect the development and communities of the understory vegetation, which would be closely related to the modification of microclimates and alterations of edaphic factors. Therefore, in order to test this hypothesis, an experiment was conducted involving different thinning intensities to investigate the changing mechanisms of the understory vegetation in the study area. The specific aims were as follows: (1) to determine the effects of different thinning intensities on the diversity, coverage, and biomass of the understory vegetation; (2) to analyze the community composition and distribution pattern of the understory vegetation impacted by the different thinning intensities; and (3) to identify the most important factors driving the community composition and distribution pattern of the understory vegetation.

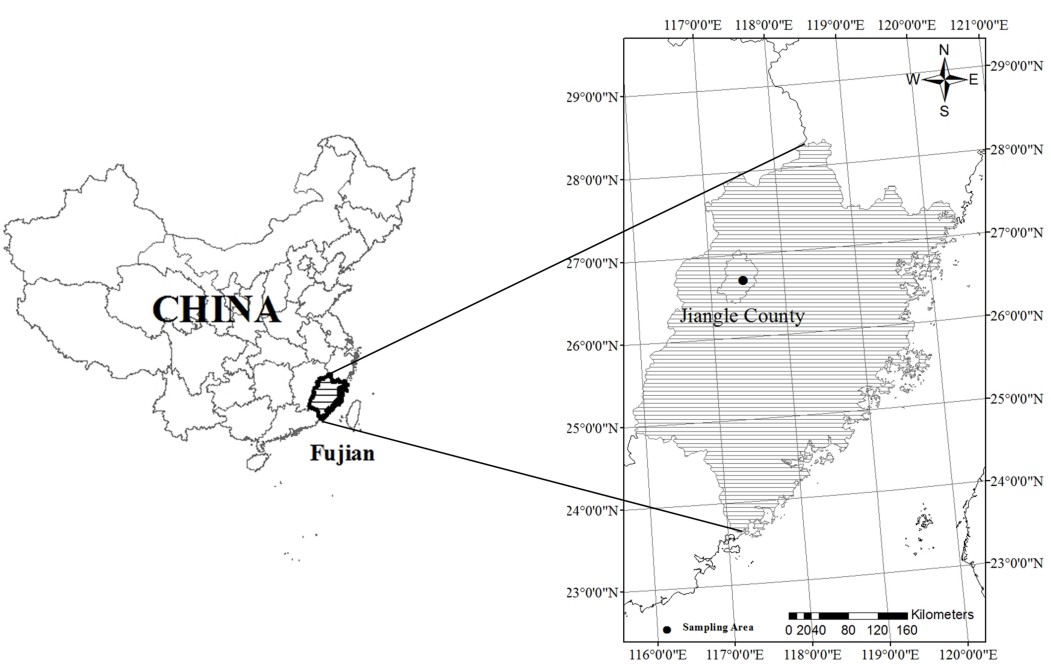

**Figure 1** **Locations of the study area.**

## MATERIAL AND METHODS

### Study area and site description

This study area was located at Jiangle forest farm in Fujian Province of Southeastern China ($26°26'-27°04'$N, $117°05'-117°40'$E), with an elevation ranging between 200 and 800 m above sea level (Fig. 1). The climate in the study area was found to be subtropical monsoon climate. The mean annual temperature was determined to be 19.5 °C, and the mean annual precipitation was 1,665 mm (data from National Meteorological Information Center, observed from 1981 to 2010, http://data.cma.cn/site/index.html). This region was dominated by low-mountain hilly terrain. The soil in the study area was identified as ultisols (locally known as red clay soil) according to the USDA Soil Taxonomy (*Li et al., 2019*). The dominant tree species in the region included Chinese fir and Masson pine (*Pinus massoniana*). The prominent understory species included *Callicarpa giraldii*, *Ficus hirta*, *Blechnum orientale*, *Stenoloma chusanum*, *Dicranopteris dichotoma*, and *Woodwardia japonica*.

The study area was dominated by a Chinese fir forest which had first been planted in 2005. The total area measured 3 ha and featured a similar site quality (Site index: 18 m, base age of 20 years). The thinning experiments were initially established by *Zhang (2015)* in May of 2013 when the trees were eight years old. The experimental process included the following steps: firstly, after surveying the actual terrain and eliminating the edge effects, the study area was divided into four experimental sites measuring 100 m × 30 m, from upslope to the downward slope; Then, four thinning treatments were conducted in the four sites, which included a control check (CK: no thinning), a low intensity thinning (LIT:

**Table 1  Basic conditions of the sample plots.**

| Plots | Area of plots (m²) | Aspect | Slope | Altitude (m) | Number of tree (Stem) | Reserved tree density (Stem ha⁻¹) | Mean DBH (cm) | Mean tree height (m) | Canopy density (%) |
|---|---|---|---|---|---|---|---|---|---|
| CK1 | 15 × 20 | 311° | 32° | 246.5 | 87 | 2,900 | 11.8 | 10.0 | 88 |
| CK2 | 15 × 20 | 310° | 31° | 221.8 | 88 | 2,933 | 12.8 | 9.9 | 89 |
| CK3 | 15 × 20 | 303° | 34° | 206.3 | 92 | 3,067 | 13.2 | 9.0 | 90 |
| LIT1 | 15 × 20 | 304° | 35° | 249.8 | 74 | 2,467 | 13.3 | 9.6 | 74 |
| LIT2 | 15 × 20 | 346° | 32° | 235.4 | 75 | 2,500 | 12.2 | 8.8 | 73 |
| LIT3 | 20 × 20 | 292° | 31° | 215.9 | 95 | 2,375 | 13.2 | 8.2 | 70 |
| MIT1 | 15 × 20 | 295° | 30° | 258.2 | 58 | 1,933 | 14.0 | 8.8 | 58 |
| MIT2 | 15 × 20 | 300° | 33° | 239.6 | 61 | 2,033 | 12.9 | 9.4 | 65 |
| MIT3 | 20 × 20 | 296° | 33° | 215.2 | 85 | 2,125 | 13.6 | 9.9 | 65 |
| HIT1 | 15 × 20 | 296° | 31° | 262.1 | 45 | 1,500 | 12.1 | 8.5 | 50 |
| HIT2 | 15 × 20 | 303° | 39° | 238.5 | 50 | 1,667 | 13.4 | 8.5 | 52 |
| HIT3 | 15 × 20 | 318° | 31° | 215.4 | 49 | 1,633 | 13.1 | 7.9 | 50 |

approximately 20%), a moderate intensity thinning (MIT: approximately 33%), and a high intensity thinning (HIT: approximately 50%) (*Ma, Li & Wang, 2007*). Before thinning, the average diameter at breast height (DBH) and tree height were 12.83 cm and 8 m, the average tree density was 2,967 stems ha⁻¹, the stand basal area was 38.00 m² ha⁻¹, the stand volume was 198.07 m³ ha⁻¹. After thinning, the average DBH and tree height were 13.08 cm and 9.1 m, the average residual density was from 1,600 to 2,967 stems ha⁻¹, the stand basal area was 29.96 m² ha⁻¹, the stand volume was 169.93 m³ ha⁻¹. Finally, three replicate plots were established at each thinned site in August of 2013. A total of 12 plots were established, and each plot was situated 5 m away from the next plot to minimize potential edge effects. The detailed information of the plots was shown in Table 1.

## Plots investigations

The surveys of the understory vegetation were conducted in July of 2015 and 2016. The investigations of the understory vegetation were performed in five 5 × 5 m subplots within each plot and five 1 × 1 m quadrats within each subplot for the shrubs and herbs, respectively, at the center and each of the four corners of the study area. Initially, the species were identified according to the information obtained from the Chinese website eFlora of China (http://frps.iplant.cn) and Plant Photo Bank of China (http://ppbc.iplant.cn). The species, number, and coverage data of all the shrubs and herbs were investigated for the purpose of estimating the species richness and vegetative cover for this study's subsequent analyses. The species richness was defined as the number of species per plot (*Widenfalk & Weslien, 2009*). The coverage was estimated as the ratio of the vertical projection of exposed leaf area to the area of each subplot. The average values of the vegetative cover in the five subplots effectively described the vegetation cover within each plot. Then, three of the five subplots (along the diagonal) were chosen to survey the understory biomass of each plot. All of the individuals were harvested (including roots) in each subplot and rapidly weighed to obtain the fresh weight data. The branches, leaves, and roots of the shrubs were

distinguished, as well as the above- and below-ground components of the herbs. At that point, all of the samples were oven-dried at 105 °C until reaching a constant weight and subsequently used to estimate the biomass per hectare of each plot. The biomass was only investigated during the 2016 survey process, considering the destruction of the understory vegetation caused by the applied whole harvesting method.

Following the random selection of three soil sampling points within each plot, the physical properties of the soil were determined depending on the operational standards (*State Forestry Bureau, 1999*). In the present study, a soil cutting ring (100 cm$^3$) was used at a depth of 20 cm. The soil moisture content (SMC) was calculated according to the equation SMC = (Ww – Wd)/Wd $\star$100%, in which Ww represented the wet weight of the soil, and Wd indicated the dry weight of the soil (oven-dried at 105 to 110 °C until reaching a constant weight). Furthermore, additional soil samples were collected from each sampling point at 0–20 cm soil layer using valve bags. These soil samples were divided into two parts: one was air-dried and sieved (less than two mm) to measure the soil chemical properties according to the method described by *Bao (2000)*, the other was stored at 4 °C immediately for measurement of microbial biomass carbon (MBC) and microbial biomass nitrogen (MBN). In order to measure the soil pH values, the soil: water suspensions (1:5) were stirred for 30 min and then measured by a pH meter (LEICI, China). The soil organic carbon (SOC) was estimated by utilizing an elemental analyzer (FLASH2000 CHNS/O, USA). In addition, the soil total nitrogen (TN) and total phosphorus (TP) contents were determined using an AA3 continuous flow analytical system (AA3, Germany) after digesting with $HClO_4$-$H_2SO_4$. The soil magnesium (Mg) and manganese (Mn) contents were measured using an atomic absorption spectrometer (TAS-990AFG, China) after digesting with $HNO_3$-$HClO_4$. The soil available nitrogen (AN) content was determined using an alkali solution diffusion method. The soil available phosphorus (AP) and available potassium (AK) contents were evaluated using spectrophotometry equipment (UV-2600, Japan) after HCl-$NH_4$F extraction, and ammonium acetate extraction, respectively. MBC and MBN in the soil were estimated by adopting the fumigation-extraction method.

## Statistical analyses

A one-way analysis of variance (ANOVA) method was adopted to assess the significant differences among the thinning intensities (four levels) on the characteristics of understory plants (species richness, coverage, and biomass), as well as the soil properties (SMC, pH, SOC, TN, TP, AN, AP, AK, Ca, Mg, Mn, MBC, and MBN). The analysis of the significant differences in the multiple comparisons among the four thinning intensities was performed using Tukey's HSD test at $P < 0.05$. A Pearson correlation analysis was used to detect the relationship between understory plant characteristics and soil properties.

The biodiversity indices (e.g., species richness) can quantify biodiversity and be comparable among different studies, but they place more emphasis on species counts rather than community structures and ecologically significant changes (*Haughian & Frego, 2016*). In contrast, matrix-based dissimilarity assessment provides compositional changes than biodiversity indices. In the present study, a principal coordinate analysis (PCoA) with the Bray-Curtis dissimilarity measurement were adopted to elucidate the dissimilarities of

the understory community compositions among the various plots (*Oksanen et al., 2013*). Furthermore, indicator species analysis (ISA) was adopted to evaluate the consistency and connection between species and treatments, i.e., such species refer to 'associates' rather than special 'indicators' (*Haughian & Frego, 2016*).

This study included a canonical correspondence analysis (CCA) process, which was adopted to demonstrate the relationships between the environmental factors and understory species compositions, as well as to detect principal environmental factors impacting the understory structures. Before the CCA, a pre-selection of the explanatory variables was implemented using a Mantel test /Partial Mantel test method. In addition, a number of the variables were converted before being used in this study's analyses. The aspect was transformed from a 0° to 360° Compass scale to a value between 0 and 1 (*Yu & Sun, 2013*). The transformation was performed by the following formula:

$$TRASP = \{1 - \cos[(\pi/180) * (aspect - 30)]\}/2.$$

The slope positions were converted to numerical values of 1, 2, and 3, which represented lower, middle, and upper slopes, respectively. In addition, the correlations of the environmental factors showed that the correlation between the SOC and TN contents, AP and AN contents, and Mg and Mn contents of the soil were high. Therefore, these soil contents were merged into C and N content (SOILCN), available nutrient content (SOILA), and Mg and Mn content (SOILM), respectively. During the final analysis process, the species matrix contained the presence and absence data of the species. The environmental matrix contained 11 variables, namely tree density, slope position, SMC, SOILCN, TP, SOILA, AK, and SOILM.

The ANOVA results with multiple comparisons, as well as the figures, were finished via Origin 2019 software. The ISA was conducted with the function of 'indval' in R 3.3.3. Then, the PCoA and CCA were performed by the statistical package of 'vegan' in R 3.3.3.

## RESULTS

### Effects of thinning intensity on the soil and understory characteristics

It was observed that the different thinning intensities had significantly influenced the SOC, AP, and AK contents, with the highest content detected in the HIT of the tested soil samples, respectively (Table 2). The SMC, pH, TN, TP, AN, Mg, Mn, MBN, and MBC were found to be comparable among the four thinning intensities, without observable variations.

The species richness, total coverage, and biomass of the understory vegetation were also found to be significantly influenced by the different thinning intensities, and both the shrubs and herbs had remarkably increased with the increasing thinning intensities. The highest values of species richness, total coverage, and biomass of the understory were detected in the HIT (Fig. 2).

### Relationships between the soil and the understory characteristics

The species richness, total coverage, and biomass of the understory vegetation were found to be strongly correlated with the tree density, which confirmed that the intensities of the
**Table 2 Soil properties among four thinning intensities of Chinese fir plantations.**

| Variations | CK | LIT | MIT | HIT | F | P |
|---|---|---|---|---|---|---|
| SMC (%) | 29.51 ± 4.64 | 30.36 ± 4.73 | 31.57 ± 5.30 | 34.19 ± 7.47 | 0.13 | 0.939 |
| pH | 4.40 ± 0.12 | 4.37 ± 0.08 | 4.34 ± 0.05 | 4.43 ± 0.09 | 0.18 | 0.906 |
| SOC (g kg$^{-1}$) | 20.71 ± 1.68b | 35.06 ± 1.43ab | 29.59 ± 3.77ab | 42.84 ± 6.17a | 6.07 | 0.019[*] |
| TN (g kg$^{-1}$) | 0.12 ± 0.02 | 0.21 ± 0.02 | 0.19 ± 0.03 | 0.28 ± 0.06 | 2.95 | 0.098 |
| TP (g kg$^{-1}$) | 0.32 ± 0.01 | 0.32 ± 0.02 | 0.30 ± 0.01 | 0.35 ± 0.01 | 1.52 | 0.282 |
| AN (mg kg$^{-1}$) | 118.72 ± 15.28 | 171.11 ± 16.62 | 146.74 ± 8.26 | 162.33 ± 20.73 | 2.10 | 0.179 |
| AP (mg kg$^{-1}$) | 1.80 ± 0.28b | 3.79 ± 0.52ab | 3.51 ± 0.25ab | 4.73 ± 0.63a | 7.33 | 0.011[*] |
| AK (mg kg$^{-1}$) | 58.09 ± 3.83ab | 51.48 ± 3.37b | 47.87 ± 3.57b | 70.44 ± 3.11a | 8.14 | 0.008[**] |
| Mg (mg kg$^{-1}$) | 69.19 ± 6.61 | 80.07 ± 6.84 | 73.18 ± 7.14 | 74.20 ± 9.16 | 0.36 | 0.785 |
| Mn (mg kg$^{-1}$) | 85.65 ± 21.63 | 122.46 ± 39.55 | 88.09 ± 23.95 | 121.19 ± 24.60 | 0.51 | 0.687 |
| MBC (mg kg$^{-1}$) | 121.07 ± 34.33 | 166.17 ± 67.15 | 221.67 ± 31.47 | 149.70 ± 46.71 | 0.81 | 0.524 |
| MBN (mg kg$^{-1}$) | 41.33 ± 5.26 | 32.70 ± 2.22 | 35.80 ± 10.25 | 95.80 ± 42.11 | 1.86 | 0.215 |

**Notes.**

Values are mean ± stand error; * indicates $P < 0.05$; ** indicates $P < 0.01$; and *** indicates $P < 0.001$; Different letters within a row indicate significant differences ($P < 0.05$) among the different treatments based on the one-way ANOVA results, followed by the Tukey HSD test results.

CK, no thinning; LIT, low intensity thinning; MIT, moderate intensity thinning; HIT, high intensity thinning; SMC, soil moisture content; SOC, soil organic carbon; TN, total nitrogen; TP, total phosphorous; AN, available nitrogen; AP, available phosphorous; AK, available potassium; MBC, microbial biomass carbon; MBN, microbial biomass nitrogen.

thinning process had significantly influenced the understory characteristics (Table 3). Furthermore, as can be seen in Table 3, it was also indicated that the understory characteristics were prominently positive relevant to the soil nutrients (for example, SOC, TN, TP, AP, and AK). However, such micro-topography factors as the aspect, slope, and slope position, as well as the soil Mg, Mn, MBC, and MBN contents, were not observed to be correlated with the understory vegetation characteristics.

## Effects of thinning intensity on the understory species compositions

In the present study, 60 species of understory plants were discovered, including 31 shrub species belonging to 19 families and 24 genera, and 29 herb species belonging to 18 families and 24 genera. Among the 29 herb species, 22 species were ferns, accounting for 76% of the herbaceous species. The dominant shrub species were *C. giraldii* and *F. hirta*. The dominant herb species were *Allantodia metteniana* and *W. japonica* (Table S1). The ISA results identified one shrub species and five herb species which could be used as conspicuous indicators of the understory communities among the four thinning intensities. All of these species were identified as the indicators in HIT, as detailed in Table 4.

The PCoA results revealed significant changes in the understory community compositions among the four thinning intensities ($P = 0.047$). The ordination showed that the CK (no thinning) was separate from both the MIT (moderate) and HIT (high), while the LIT (low) and MIT (moderate) had clustered closely with each other (Fig. 3), which illustrated that understory communities were significantly different between CK and HIT.

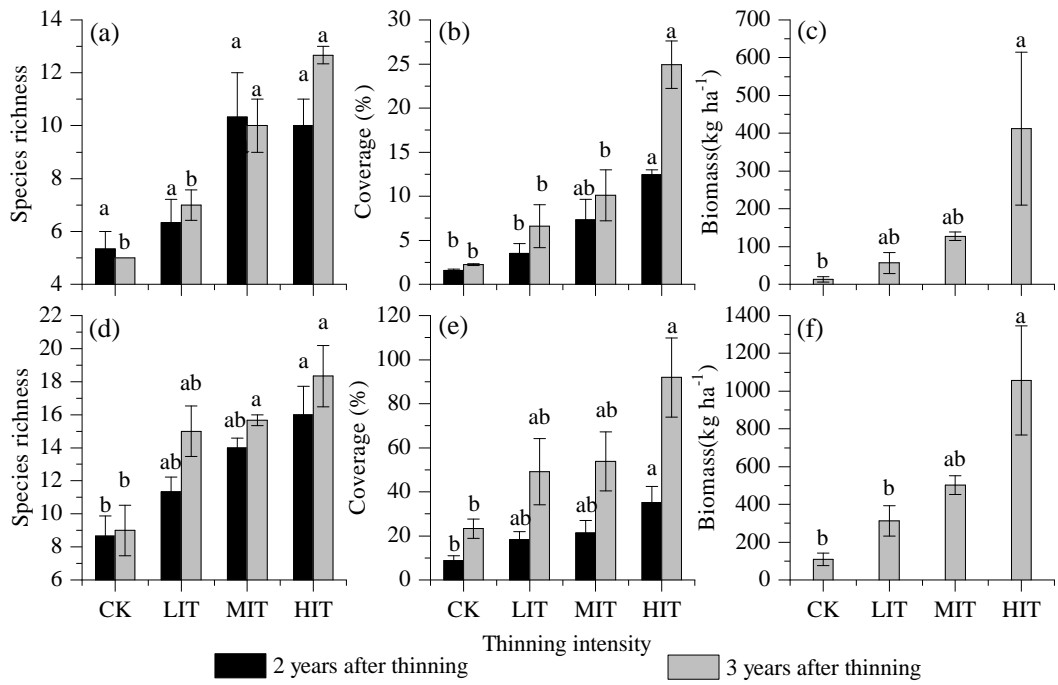

**Figure 2** **Species richness, vegetation cover, and biomass among the four thinning intensities in the shrub and herb layers.** (A) Species richness in shrub layers. (B) Vegetation cover in shrub layers. (C) Biomass in shrub layers. (D) Species richness in herb layers. (E) Vegetation cover in herb layers. (F) Biomass in herb layers. Values are mean ± stand error. Different letters indicate significant differences ($P < 0.05$) among the four different thinning treatments based on the one-way ANOVA results, followed by the Tukey HSD test results. CK, no thinning; LIT, low intensity thinning; MIT, moderate intensity thinning; HIT, high intensity thinning.

## Relationships between the understory communities and the environmental factors

The Mantel test results revealed that the understory species compositions were principally affected by the tree densities, slope positions, C and N contents of the soil, and the Mg and Mn contents of the soil, as illustrated in Table 5. The results of the canonical correspondence analysis (CCA) showed that these variables explained a total of 32.4% of the variation in the species distribution among plots, with Axis 1 explaining 17.4% and Axis 2 explaining 15.0% of the total variation (Fig. 4). The ordination plot revealed that the tree densities were the prominent factor which contributed to the understory community distributions among different thinning intensities, as well as SOILCN contributed to it among thinning intensities (Fig. 4). When controlling other environmental factors, slope position showed the strongest effect on understory community composition ($P = 0.001$), while other soil nutrient contents (C and N, Mg and Mn) showed less but significant influences (Table 5).

**Table 3   Correlations between understory characteristics and environmental factors.**

| | 2 years after thinning | | | | 3 years after thinning | | | | | |
| | Shrub layers | | Herb layers | | Shrub layers | | | Herb layers | | |
| | Species richness | Cover | Species richness | Cover | Species richness | Cover | Biomass | Species richness | Cover | Biomass |
|---|---|---|---|---|---|---|---|---|---|---|
| SMC (%) | 0.030* | 0.253 | 0.686 | 0.401 | 0.614 | 0.601 | 0.948 | 0.300 | 0.429 | 0.718 |
| pH | 0.232 | 0.807 | 0.358 | 0.964 | 0.983 | 0.813 | 0.354 | 0.485 | 0.716 | 0.593 |
| SOC (g kg⁻¹) | 0.184 | 0.014* | 0.040* | 0.066 | 0.044* | 0.004** | 0.001*** | 0.062 | 0.092 | 0.000*** |
| TN (g kg⁻¹) | 0.208 | 0.021* | 0.092 | 0.176 | 0.068 | 0.014* | 0.001*** | 0.142 | 0.267 | 0.000*** |
| TP (g kg⁻¹) | 0.657 | 0.113 | 0.645 | 0.472 | 0.090 | 0.043* | 0.091 | 0.098 | 0.186 | 0.067 |
| AN (mg kg⁻¹) | 0.408 | 0.181 | 0.807 | 0.770 | 0.273 | 0.229 | 0.198 | 0.066 | 0.641 | 0.127 |
| AP (mg kg⁻¹) | 0.221 | 0.011* | 0.065 | 0.197 | 0.002** | 0.004** | 0.007** | 0.004** | 0.110 | 0.001*** |
| AK (mg kg⁻¹) | 0.809 | 0.063 | 0.747 | 0.282 | 0.178 | 0.034* | 0.205 | 0.392 | 0.184 | 0.168 |
| Mg (mg kg⁻¹) | 0.897 | 0.729 | 0.877 | 0.418 | 0.674 | 0.706 | 0.332 | 0.476 | 0.167 | 0.527 |
| Mn (mg kg⁻¹) | 0.528 | 0.613 | 0.820 | 0.648 | 0.419 | 0.407 | 0.955 | 0.329 | 0.311 | 0.905 |
| MBC (mg kg⁻¹) | 0.717 | 0.746 | 0.681 | 0.690 | 0.457 | 0.931 | 0.597 | 0.119 | 0.747 | 0.867 |
| MBN (mg kg⁻¹) | 0.180 | 0.170 | 0.687 | 0.366 | 0.105 | 0.259 | 0.744 | 0.083 | 0.366 | 0.438 |
| TD (Stem ha⁻¹) | 0.005** | 0.000*** | 0.000*** | 0.005** | 0.000*** | 0.000*** | 0.011* | 0.001*** | 0.007** | 0.000*** |
| Aspect | 0.836 | 0.752 | 0.611 | 0.401 | 0.631 | 0.972 | 0.534 | 0.793 | 0.233 | 0.587 |
| Slope (°) | 0.627 | 0.639 | 0.735 | 0.760 | 0.570 | 0.925 | 0.425 | 0.369 | 0.814 | 0.659 |
| Slope position | 0.341 | 0.975 | 0.845 | 0.991 | 0.755 | 0.837 | 0.523 | 1.000 | 0.608 | 0.548 |

Notes.
*$P < 0.05$.
**$P < 0.01$.
***$P < 0.001$.
TD, Tree density. Other descriptions as in Table 2.

**Table 4   Indicator species of different treatments based on indicator species analysis (ISA) results.**

| Layers | Treatment | Species | Indicator values | P |
|---|---|---|---|---|
| Shrub | HIT | *C. giraldii* | 0.511 | 0.049 |
| Herb | HIT | *S. viridis* | 1.000 | 0.013 |
| | | *D. indica* | 1.000 | 0.023 |
| | | *P. glanduligera* | 0.825 | 0.031 |
| | | *A. japonica* | 0.553 | 0.030 |
| | | *W. japonica* | 0.385 | 0.047 |

Notes.
HIT, high intensity thinning.

## DISCUSSION

Our research highlighted that the thinning measures significantly improved the occurrence, growth rates, diversity and community formation of the understory vegetation, as well as the soil properties of Chinese fir plantations in Southeastern China. These results had great significance of the forest management in Chinese fir plantations. The responses of understory vegetation to thinning intensities were primarily associated with the canopy density and microenvironment in the stand.

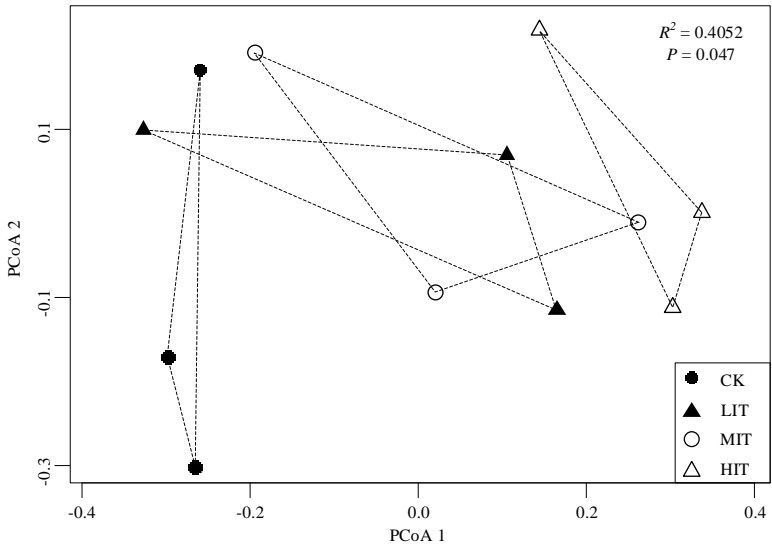

**Figure 3** **Principal coordinate analysis (PCoA) of the dissimilarities in the understory communities among the different thinning intensities.** CK, no thinning; LIT, low intensity thinning; MIT, moderate intensity thinning; HIT, high intensity thinning.

**Table 5** **Relationships between the understory communities and the environmental factors as revealed by the Mantel test results.**

| Explanatory variables | $R^2$ | $P$ |
|---|---|---|
| TD (Stem ha$^{-1}$) | 0.6752 | 0.005[**] |
| Aspect | 0.0322 | 0.862 |
| Slope (°) | 0.0468 | 0.796 |
| Slope Position | 0.8001 | 0.001[***] |
| SMC (%) | 0.1605 | 0.490 |
| pH | 0.0807 | 0.724 |
| SOILCN (g kg$^{-1}$) | 0.6009 | 0.029[*] |
| TP (mg kg$^{-1}$) | 0.3753 | 0.173 |
| SOILA (mg kg$^{-1}$) | 0.1295 | 0.588 |
| AK (mg kg$^{-1}$) | 0.4308 | 0.104 |
| SOILM (mg kg$^{-1}$) | 0.6764 | 0.005[**] |
| MBC (mg kg$^{-1}$) | 0.0635 | 0.789 |
| MBN (mg kg$^{-1}$) | 0.1053 | 0.536 |

**Notes.**
[*]$P < 0.05$.
[**]$P < 0.01$.
[***]$P < 0.001$.
SOILCN, Soil C and N content; SOILA, Soil available nutrient content; SOILM, Soil Mg and Mn content.
Other descriptions as in Table 2.

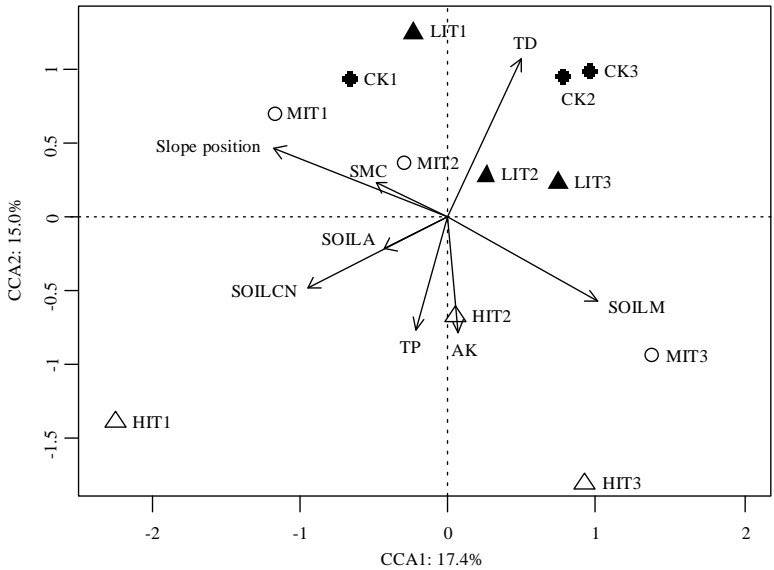

**Figure 4** Canonical correspondence analysis (CCA) results of the dissimilarities in the understory communities among the different thinning intensities. CK, no thinning; LIT, low intensity thinning; MIT, moderate intensity thinning; HIT, high intensity thinning. Other descriptions as in Tables 2 and 5.

In the present study, it was found that the species richness, coverage, and biomass of the undergrowing shrubs and herbs had increased significantly with the increasing thinning intensities (Fig. 2). These findings had supported this study's hypothesis. The results were not found to be surprising since it has been verified that thinning practices would potentially improve the spaces and light transmittance abilities in densely treed areas by reducing the canopy densities. As a result, the nutrient areas and growth spaces of the understory vegetation were enhanced (*Sheng, 2001a*; *Ma et al., 2007*; *Ares, Neill & Puettmann, 2010*; *Cañellas et al., 2004*). Furthermore, the results obtained in this study were found to be in good agreement with those of previous related studies, in which thinning had enhanced the diversity, coverage, and biomass of undergrowth vegetation (*Trentini et al., 2017*; *Zhou et al., 2016*; *Dang et al., 2018*). For example, *Zhou et al. (2016)* investigated the understory vegetation of Chinese fir plantations in Southern China and found the biodiversity and biomass of the undergrowth vegetation had been significantly increased after pre-commercial thinning. This study's results further confirmed that the species richness, coverage, and biomass of the undergrowth vegetation significantly increased with the increasing of the thinning intensities. However, some of the previous related researches have suggested that thinning practices had little or negative impacts on undergrowth (*Cheng et al., 2014*; *Cheng et al., 2017*). These contradictory results with regard to thinning effects on understory vegetation growth rates and biodiversity may have resulted from differences in forest stand circumstances, thinning intensities, plant community structures, and the stand ages of the forests (*Brett & Moniquee, 2009*; *Cheng et al., 2017*; *Juodvalkis, Kairiukstis & Vasiliauskas, 2005*). In the area examined in the present study, a high initial planting density had caused insufficient light transmittance to the undergrowth, as well

as negatively affected the availability of soil nutrients for the understory vegetation. These factors may have led to the weak development of undergrowth vegetation (*Sheng, 2001b*). Therefore, when the undergrowth vegetation acquired enough light, nutrients, and growth space after thinning was implemented, the development of the understory plants had increased (*Sheng, 2001b*; *Cheng et al., 2017*).

The results obtained in this study also showed that thinning had significantly affected the SOC, AP, and AK contents in the study area, as detailed in Table 2. The results showed that thinning measures improved the soil nutrients contents, which was consistent with the results of the *Dang et al. (2018)*. In addition, the further analyses of the correlations between the understory vegetation and the soil properties indicated that the species richness, as well as the coverage and biomass of the understory vegetation, had significantly correlated with the soil nutrient contents, as illustrated in Table 3. On one hand, the improvements in light transmittance and temperature had resulted in higher decomposition rates after thinning. On the other hand, it was found that the improved development of the understory plants after thinning had led to an increasing proportion of rapidly decomposable litter, which could effectively supplement the available nutrient content (for example, SOC, AN, AP, and AK) input to the soil (*He & Barclay, 2000*; *Teste, Lieffers & Strelkov, 2012*; *Yao, Sheng & Xiong, 1991*).

The results of this study indicated that the community compositions of the understory vegetation had significantly differed among the four thinning intensities, and the variations between the CK and HIT sections were most notable. Meanwhile, it was found that the LIT and MIT sections were highly similar (Fig. 3). Tree densities could be one of the significant factors which influenced the community compositions of the understory vegetation. Several previous related studies had indicated that the tree layers were important factors which also indirectly influenced the understory plants through the alteration of site factors such as litter mass and canopy openness (*Brosofske, Chen & Crow, 2001*; *Wulf & Naaf, 2009*; *Yu & Sun, 2013*). However, it was also indicated that the understory community similarity characteristics were high among the different thinning intensities, indicating that thinning does not necessarily influence the community compositions of the understory plants since there were no extreme changes in forest types and environments among the thinning intensities (*He et al., 2019*). In the present study, the significance of the tree density in the understory vegetation distributions suggested there were strong controls exerted on the understory communities by the different thinning intensities. The variations of the understory community compositions among the four examined thinning intensities were found to be caused by changes in the light transmittance within the forest, which is beneficial to the growth of some light-demanding species (*Cañellas et al., 2004*). That is, there might be only a small number of shade-tolerant understory species in the un-thinned stands, which led to a very simple understory species composition. However, most pre-established species increased as the canopy thinning, especially light-favoring pioneer species. Subsequently, the forest had formed multi-storied communities due to the reductions in both tree densities and light deficiency (*Zhou et al., 2016*). Some light-demanding species were identified as significant indicators in the HIT section, such as *C. giraldii, P. glanduligera,* and *A. japonica.*

It was also determined by this study's results that the soil properties were demonstrated as the primary factors affecting the community compositions of understory vegetation among all of the different thinning intensities (Fig. 4; Table 5). Previous studies had illustrated the soil nutrient contents were important factors which influenced the community compositions of the understory vegetation (Ploughe & Dukes, 2019; Zarfos et al., 2019). For example, Zarfos et al. (2019) demonstrated that the community compositions displayed significant changes with different gradients of soil acidity, base cation availability, and carbon-to-nitrogen ratios (C:N). Furthermore, Legare et al. (2001) found that the canopy types affected the understory compositions through impacting nutrient availability. However, other related studies had reflected that the levels of soil fertility did not play significant roles in the characteristics of understory communities (Yu & Sun, 2013). In the current study, it was determined that the soil C and N content (inferred by the data of the soil organic carbon and nitrogen) and soil Mg and Mn contents had appeared to play significant roles in the understory community characteristics in the study area.

Most importantly, it was unexpectedly found that slope position was an important topographical factor contributed to community compositions of understory vegetation (Fig. 4; Table 5). Favorable positions were determined to be the primary drivers of the observed differences in community compositions of understory vegetation among the different thinning intensities. Generally speaking, on such small scales as stands and communities, the overstory structural, topographical, and edaphic factors were generally considered to be the primary drivers of the observed differences in understory community compositions (Van Couwenberghe et al., 2010; Siefert et al., 2012). Slope position, as an important topographical factor, determined the heterogeneity of the microenvironment of the local habitat through controlling such conditions as light, temperature, and moisture in the forest (Liang et al., 2017), and thereby affected soil structure and development (Seibert, Stendahl & Sørensen, 2007; Mohammadi et al., 2016), soil nutrients redistribution and availability by gravity and hydraulic power (Fisk, Schmidt & Seastedt, 1998; Xiao et al., 2019). As a result, slope position would influence the growth, diversity, distribution, and community composition of vegetation (Zhou, Liu & Hou, 2009; Zhang et al., 2011; Lei et al., 2018). Previous studies have shown that topographical factors play a part in determining the differences in the community compositions of understory vegetation (Dai, Hirabuki & Mochida, 2010; Yang, Da & You, 2005; Yang & Da, 2006; Zhang et al., 2011). These topographical factors include altitude gradient (Coll et al., 2011; Saima et al., 2018) and aspect (Bennie et al., 2006; Yu & Sun, 2013; Yu et al., 2013). In this study, the slope positions were detected as the most significantly topographical factor affecting the community compositions of the understory vegetation, which had been rarely mentioned in previous related studies. Our study area is located in the hilly areas of Southern China, where the topography is heterogeneous, and thereby the micro-topographic factors, such as slope position, may strongly affect the ecological environments of the local habitat, as well as the soil and vegetation. Therefore, our results were instructive for forestry researchers to understand that slope position should be considered in the management and further researches of such plantation ecosystems.

Although this study was conducted in a short-term (three years) after thinning, the results were instructive for forestry researchers and managers to understand the effects of the different thinning intensities on the development and community compositions of the understory vegetation, which provided a clear understanding of the mechanisms of thinning management for Chinese fir plantations in Southeastern China, and suggested that high intensity thinning could potentially improve the development of the understory plants. In addition, it can be expected that in other forest ecosystems with high stand density in the world, reasonable thinning measures could improve the development of understory vegetation, increase the biodiversity, coverage and biomass of undergrowth, and promote community formation of understory plants, which enhance the productivity and sustainable management of forests. Furthermore, in order to objectively assess the effects of thinning intensities on the understory vegetation, and provide more reliable long-term forest management measures, it is recommended that future studies should focus on long-term continuous observations and the analysis of any observed dynamic changes which may occur over time.

## CONCLUSIONS

The results of this study strongly indicated that the thinning of Chinese fir plantations significantly influenced the development, diversity, and community composition of the understory vegetation of Chinese fir plantations in this study area, as well as the properties of the soil. The species richness, total coverage, and biomass of the understory plants were observed to significantly increase with the increasing thinning intensities. It was also found that thinning measures had prominently influenced the soil nutrients (for example, SOC, AP, and AK). The community compositions of the understory vegetation were observed to be significantly different among four thinning intensities. This was found to be especially obvious between CK (control) and HIT (high intensity). In addition, the growth of the understory vegetation was significantly correlated with the soil nutrient contents (for example, SOC, TN, TP, AP, and AK), and the community compositions of the understory vegetation were prominently driven by tree densities, slope positions, soil C and N contents, and soil Mg and Mn contents. These findings suggested that high intensity thinning (approximately 50%) should be conducted in Chinese fir plantations, which could potentially improve the development of the understory plants of Chinese fir plantations in Southeastern China and expected in other forest ecosystems with high stand density in the world.

## ACKNOWLEDGEMENTS

We appreciate Peng Zhang who established the thinning experiments. We also acknowledge the use of the facilities at the Centre Experiment and Teaching, College of Forestry, Beijing Forestry University.

### Funding

This work was supported by the National Natural Science Foundation of China: Study on crown models for *Larix olgensis* based on tree growth (No. 31870620), the National Key Research & Development (R&D) Program of China: tending regeneration technique of spruce-fir forest (2017YFC050410101) and the Short-term International Student Program for Postgraduates of Forestry First-Class Discipline (2019XKJS0501). The funders had no role in study design, data collection and analysis, decision to publish, or preparation of the manuscript.

### Grant Disclosures

The following grant information was disclosed by the authors:
National Natural Science Foundation of China: Study on crown models for *Larix olgensis* based on tree growth: No. 31870620.
National Key Research & Development (R&D) Program of China: tending regeneration technique of spruce-fir forest: 2017YFC050410101.
Short-term International Student Program for Postgraduates of Forestry First-Class Discipline: 2019XKJS0501.

### Competing Interests

The authors declare there are no competing interests.

### Author Contributions

- Xuelei Xu conceived and designed the experiments, performed the experiments, analyzed the data, prepared figures and/or tables, authored or reviewed drafts of the paper, and approved the final draft.
- Xinjie Wang conceived and designed the experiments, authored or reviewed drafts of the paper, and approved the final draft.
- Yang Hu performed the experiments, authored or reviewed drafts of the paper, and approved the final draft.
- Ping Wang analyzed the data, prepared figures and/or tables, and approved the final draft.
- Sajjad Saeed analyzed the data, authored or reviewed drafts of the paper, and approved the final draft.
- Yujun Sun conceived and designed the experiments, authored or reviewed drafts of the paper, and approved the final draft.

### Data Availability

The raw measurements are available as a Supplementary File.

### Supplemental Information

Supplemental information for this article can be found online at http://dx.doi.org/10.7717/peerj.8536#supplemental-information.

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
