# Peer review of "Short-term effects of thinning on the development and communities of understory vegetation of Chinese fir plantations in Southeastern China"

_PeerJ, doi:10.7717/peerj.8536_

## Round 0.1 · original submission · Major Revisions

Please take carefully in account the suggestions of the reviewers.

Reviewer 1 ·

Basic reporting

The submitted manuscript entitled Short-term effects of thinning on the development and communities of understory vegetation of Chinese fir plantations in Southeastern China (#41940) presents results of experimental study on the effect of different intensity of thinning of young Cunningamia lanceolata trees on understorey vegetation.
The article is understanding and easy for following. Its structure is consistent with the PeerJ journal standards. The introduction illustrates well the importance of the research problem. However, please note that effect of tree density and thinning on understory vegetation were many times studied outside of China and obtained by Authors results are consistent with them. Hence the study topic is not very novel. The Authors mainly refer to publications from China, not referring to many other similar research conducted in other parts of the world i.e. Torras and Saura (2008), Brunet et al. (2010), and Lindh and Muir (2004) and literature cited therein. Some other studies suggests that soil conditions have higher influence on understorey vegetation that tree stand character i.e. Piwczyński et al. (2016).
Brunet J, Fritz Ö, Richnau G (2010) Biodiversity in European beech forests – a review with recommendations for sustainable forest management. Ecol Bull 53:77–94
Lindh BC, Muir PS (2004) Understory vegetation in young Douglas-fir forests: does thinning help restore old-growth composition? For Ecol Manage 192:285–296. doi: 10.1016/j.foreco.2004.01.018
Piwczyński M, Puchałka R, Ulrich W (2016) Influence of tree plantations on the phylogenetic structure of understory plant communities. For Ecol Manage 376:231–237. doi: doi.org/10.1016/j.foreco.2016.06.011
Torras O, Saura S (2008) Effects of silvicultural treatments on forest biodiversity indicators in the Mediterranean. For Ecol Manage 255:3322–3330. doi: 10.1016/j.foreco.2008.02.013


The tables, figures and its captions and supplementary materials are well prepared and do not raise objections.

Experimental design

The study hypothesis and goals are clearly presented and well discussed with obtained results and other research. The authors analysed in detail species composition and biomass of scrubs and herbs in each treatment and soil chemistry. They also consider the terrain conditions which may have influence on character of understory vegetation. The statistics are well chosen and properly interpreted. The documentary data together with used R script has attached to the manuscript as supplement.

Validity of the findings

Replication experiments is low, but it may be justified by a large amount of field and laboratory work. The research project was correctly designed and conducted with high technical and ethical standards.
The conclusions are appropriately stated, connected to the study aims and hypothesis and supported by the results. The authors should consider whether to extend the introduction and discussion so that the work is less local. In my opinion, the results are universal enough that similar patterns maybe expected in other areas of the world. Some fragments of Discussion should be better supported by literature references.
Additionally, before publishing in PeerJ journal some fragments of manuscript should be corrected.

Specific comments:
Authority on Latin binomial should be provided after each common name the first time referred to in the text.
The tables should be self-explanatory.
L.47. ‘Cunninghamia lanceolate’ -> Cunninghamia lanceolata
L.60, L.133, L.292. Ma et al, 2007 -> in references are listed two papers Ma et al, 2007
L.91. ‘Traditionally speaking’ -> Generally speaking?
L.117-119. The Authors should complete some information about source of climatic data and provide reference to soil classification.
L.148-149. ‘the species were identified according to the information obtained from the Chinese website eFlora (www.eflora.cn).’ -> Unfortunately, I cannot connect with this website address. The browser redirects me to the following website http://www.efloras.org/flora_page.aspx?flora_id=2 with electronic version and pdfs with volumes of Flora of China. If I was redirected correctly, I suggest quoting selected volumes of Flora of China (pdf/printed version) here.
Anyway, this sentence must be changed, i.e. ‘The vascular plants species were identified according (citation)’.
L.153. How vegetation cover was measured. Please specify it.
L.277-278. The abbreviations of treatments should be given.
L.314-316. This fragment should be rewritten, because in current form is difficult to understand.
L.339-348. This fragment should be discussed with literature references.
L.369. Siefert et al., 2012 -> lack in references
L.379. Seibert et al., 2007-> lack in references
L.371. Yang and Da, 2006; Yang and Da, 2005; -> Yang and Da, 2005, 2006;
L.460, L.516, L.528. Chinese text in reference should be converted to English
L.468. (2016b) -> (2016)
L.442, L.458, L.470, L.489. L.490, L.503, L.535, L.575. Lack of colon (:) or/and volume, issue or pages numbers
L.568. Probably invalid reference
Moreover, full journal titles should be given; year of publication should be given without quotes; author’s manes, year of publication and volume number should be bolded; species Latin names should be written with italics.

Additional comments

Reading the manuscript in its current form, I get the impression that it is addressed to a very narrow audience. The authors could present their results as more universal, and not only relating to the plant biodiversity of C. lanceolata plantations in E China. This may be beneficial to the future citation of the article.
Kind regards

Reviewer 2 ·

Basic reporting

The paper presents the effects of thinning of different intensities on understory and soil properties in Cunninghamia lanceolata plantations located in China. The study is interesting and, generally, well designed. Moreover, the results can potentially be important for practical foresters with regard to management of relatively extensive areas overplanted with Cunninghamia lanceolata plantations. In the research interdisciplinary approached was applied, which increases the scientific value of the study.

However, in my opinion, before the possible publication in PeerJ the paper should be amended with regard to the points listed in the section "General comments for the author".

Experimental design

Please see the general comment above.

Validity of the findings

Please see the general comment above.

Additional comments

Major points:
1. The suggested by the Authors effect of thinning intensity on growth of the overstory species (Cunninghamia lanceolata) should be substantially rethought. In fact, in my opinion the suggested effects (e.g. l. 275-278, 284, 394, 400) are unfounded by the obtained results in this study. In the present article version, the results (Fig. 5) show only that the thinning executed in 2013 was done with regard to smaller trees mainly (in fact, if the highest/thickest trees were cut the results would be opposite). Thus, the presented results do not prove any effects of thinning intensity on growth dynamics of the overstory species. Besides, I suppose that differences of mean DBH values showed in Figure 5 are insignificant between the studied variants, which were not considered by the Authors (why?). Thus, I suggest rethinking and improving this point of the study. Alternatively, the issue could be skipped in this paper in favor of another one (see the next point), which would not probably negatively influence achieving the aims of the study.
2. Discussion should be partly rethought and rewritten. There are many direct references in this section to results presented in tables in figures, which make sentences alike Results section rather, but not Discussion. Besides, the novelty of the obtained results should be stronger highlighted in this section to a reader. The Authors state several times in Discussion that the obtained results confirm previous studies. After that, a reader could have an impression that the only real novelty of the paper is the found effect of topography on the studied issues (l. 373-376). Thus, I suggest more focusing on this part of the study instead of examining controversial issues related to growth dynamics of the overstory species.



Moderate points:
1. l. 91-97: The logic of this lines should be changed. You refer here to studies conducted with regard to different tree species which clearly varied in ecology (Pinus, Pseudotsuga species etc.), and subsequently also their effects on understory. Thus, it is not surprised that the results of the previous studies are opposite. The sentence in l. 95-96 suggests that you potentially aim at showing “universal and unified” effects of thinning on understory characteristics. However, it is possible only with regard to one species or group of plant taxa, which are of similar ecological characteristics (light requirements, growth dynamics etc.), and subsequently similar effects on undergrowth. Thus, l. 91-97 should be corrected. You could start with showing the effects of thinning practices on understory in global approach, but next you should be more specific, and move exactly to the investigated species.
2. Subtitles of the Result section should be shorter, e.g. “Effects of the different thinning intensities on the soil properties and understory vegetation characteristics” could be changed for “Effects of thinning intensity on the soil and understory characteristics”.
3. With regard to plant nutrition Mg is macronutrient. I do not understand why Mg was treated, together with Mn (e.g. l. 41), as microelement throughout the paper.

Minor points:
l. 26: Following other acronyms used in the paper (LIT etc.) it is strange for a reader that CK, but not CS, is used here and throughout the article for “control section”.
l. 47: Cunninghamia lanceolata – “a” not “e”
l. 104: The end of this sentences which is placed in this line should be corrected.
l .107: change “To” for “to”
l. 110: You probably mean “most important factors”.
l. 119-120: I suggest including Soil Taxonomy in References
l. 123: The use of “respectively” is unclear in this sentence.
l. 124: delete “observed to be”
l. 125: Please say some more about the site index. E.g. were its values calculated for the age of 100 years? Is SI 18 m low or high for Chinese conditions and for the studied species?
l. 133: et al., 2007
l. 133-140: The mean values given in these lines should be followed with ±SD or SE values, when relevant.
l. 169: I recommend deleting “ground”, because it can mislead a reader. In fact, soil samples were not grounded before sieving, but aggregates were separated for +-single grains.
l. 171: “w/v” is not necessary here
l. 174, 176, 179, 180/81, 215, 216, 217 (and wherever else relevant): replace “content levels” with “contents”
l. 180-182: In the previous paragraph the methodology of collecting of soil samples for this part of the study should be described.
l. 189: Which kind of correlation was applied (Pearson; Spearman…)?
l. 206: replace “as well as detect” with “as well as to detect”
l. 228: Start the sentence just with “It was observed/stated that…”
l. 229: You should state clear what kind of “values” do you mean in this sentence. In fact, you say here (and also in other parts of the paper, e.g. l. 234) about “the strongest relation”. However, if you want to write about “values” of sth, you must specifically define what you mean.
l. 241-242: It should be clearly stated which relations you specifically mean here.
l. 265: delete “in this study”; I recommend replacing “confirmed” with e.g. “revealed”
l. 266: replace “C-N content levels” with “C and N contents” – here, but also in Table 6 and wherever relevant
l. l. 268: delete “had successfully”
l. 268/69: replace: “variations in the species distributions” with “variation in the species distribution” and “the data variations” with “the total variation”.
l. 270” change “plotting” for “plot”
l. 285: “opening (…) microenvironment” ? What do you mean?
l. 307: replace “affecting” with “affected”
l. 324: You mean input to the soil.
l. 355: C:N
l. 386: “access” – do you mean “assess”?
l. “growth” is too general. Say precisely which parameters you mean here to be correlated to soil nutrient contents.


Figure 3:
I think that the asterix (*) is not necessary in the figure.
I recommend inserting a label into the figure explaining clearly which signs are related to particular thinning variants. At the present version of the figure it could be problematic to distinguish LIT vs. MIT.

Table 1:
In the last column you probably mean canopy closure. If so, the column header should be changed and values changed for %.
The additional column should be inserted showing number of trees which were measured

Table 2:
l. 2: delete “were observed”
To be consequent the acronyms of thinning variants (CK, LIT etc.) should be also described in the table foot.

Table 3:
Replace “Correlation analysis between” with “Correlations between”
l. 1: At the end it should be stated, e.g: “Other descriptions as in table/figure x” The same statement is probably also necessary with regard to other tables/figures.

Table 5 and l. 260-262.
The results of this analysis are not significant, thus I suggest removing this. Alternatively, the Authors could try to motivate in the paper that the results prove sth new to the study. At the present form they do not.

Table 6:
The table’s title does not strictly reflect the table’s content. Not only soil characteristics are shown in the table.

---

## Round 0.2 · Minor Revisions

Please follow the suggestions of Reviewer 2

Reviewer 1 ·

Basic reporting

The revised manuscript entitled Short-term effects of thinning on the development and communities of understory vegetation of Chinese fir plantations in Southeastern China (#41940) presents well planned and well described results of experimental study on the effect of different intensity of thinning of young Cunningamia lanceolata trees on understorey vegetation. The manuscript, figures, tables and supplementary materials does not raise any methodological reservations. I am also fully satisfied with the authors' replies and provided corrections.

Experimental design

Original research with clearly defined their goal. Research issues relevant and on time.The statistics are chosen correctly and well described. Scientific data and calculation script well documented in supplementary materials. The authors introduced all necessary corrections.

Validity of the findings

The article fill a significant gap in knowledge on biodiversity of understorey vegetation in different forest thinning / management. The obtained results may have practical applications for sustainable forestry and nature conservation.
The conclusions are correct and linked with conducted research. All underlying data have been provided; they are robust, statistically sound, & controlled.

The manuscript after some corrections in quoting will be suitable to publishing in PeerJ journal.
The corrections are needed in these fragments:
L.96 (Xiong et al., 1995; Wang et al., 2010; Zhou et al., 2016; Wang et al., 2019).
L.295 (Cheng et al., 2017; Cheng et al., 2014)
L.347 Legare et al. (Legare et al., 2001)

Kind regards

Reviewer 2 ·

Basic reporting

The paper was substantially amended by the Authors following carefully my suggestions. Consequently, I recommend publishing the paper in PeerJ after minor revision only - after considering by the Authors the two still remaining points:
1. Following my suggestion, the Authors skipped in the study the issue of the effects of thinning intensity on growth dynamics of the overstory species, which was beneficial for the paper. However, I expected that the effects of topography would be treated with more detail instead. Unfortunately, the Authors inserted only some words on this topic, which, in my opinion was not sufficient. Thus, the results referring to slope aspect still should be clearer described and discussed. In fact, in the present version of the paper a reader is informed only that topography was found to be important for the study, but the nature of this effect was not described in detail.
2. With regard to the column referring to the canopy density in the Table 1 I suggest inserting the unit (%) only in the column head.

Experimental design

The study was designed properly.

Validity of the findings

The paper is interesting and shows some novel findings, which could be useful also for practical foresters.

Additional comments

The paper was substantially amended. I suggest the Authors considering the two remaining points listed above, and I wish them to successfully publish it in the journal.

---

## Round 0.3 · accepted · Accept

Dear Dr. Xu,

Your article has been accepted, thank you for your submission.

Reviewer 2 ·

Basic reporting

After correcting by the Authors the drawbacks, now I can suggest publishing the paper in PeerJ.

Experimental design

The study was designed properly.

Validity of the findings

The paper is interesting and shows some novel findings, which could be useful also for practical foresters.

Additional comments

After correcting by the Authors the drawbacks, now I can suggest publishing the paper in PeerJ.